

# High humidity tandem differential mobility analyzer for accurate determination of aerosol hygroscopic growth, microstructure and activity coefficients over a wide range of relative humidity

5     **Eugene F. Mikhailov[1,2] and Sergey S. Vlasenko[2]**

[1]Multiphase Chemistry and Biogeochemistry Departments, Max Planck Institute for Chemistry, 55020 Mainz, Germany.

[2]St. Petersburg State University, 7/9 Universitetskaya nab, St. Petersburg, 199034, Russia

10     *Correspondence to* : E. F. Mikhailov (eugene.mikhailov@spbu.ru)

**Abstract.** Interactions with water are crucial for the properties, transformation and climate effects of atmospheric aerosols. Here we present high humidity tandem differential hygroscopicity analyzer (HHTDMA) and a new method to measure the hygroscopic growth of aerosol particles with *in-situ* 15  restructuring to minimize the influence of particle shape. With this approach, growth factors can be measured with an uncertainty 0.3–0.9 % over a relative humidity (RH) range of 2–99.6 % and with an RH measurement accuracy better than 0.4 %.

    The HHTDMA instrument can be used in hydration, dehydration and restructuring modes of operation. The restructuring mode allows to investigate the effects of drying conditions on the initial 20  microstructure of aerosol particles and specified the optimal parameters that provide their rearrangements into compact structures with near-spherical shape. These optimal parameters were then used in hygroscopic growth experiments by combining restructuring mode with conventional hydration or dehydration mode. The tandem of two modes allowed us to measure the particle growth factors with high precision as well as to determine the thickness of the water adsorption layer on the surface of 25  compact crystalline particles.

    To verify HHTDMA instrument we compared the measured ammonium sulfate growth factors with these obtained from E-AIM-based Köhler model. Averaged over the range of 38–96 % RH, the mean relative deviations between measurement and model results is less than 0.5 %.

We demonstrate this precision by presenting data for glucose for which bulk thermodynamic 30  coefficients are available. The HHTDMA-derived activity coefficients of water and glucose were obtained for both dilute and supersaturated solutions and are in a good agreement with these reported in literature. Averaged deviation between the measured activity coefficients and these obtained by bulk method is less than 4 %. For dilute solution in water activity range of 0.98 - 0.99 the hygroscopicity





parameter of glucose and molal osmotic coefficient were obtained with uncertainty of 0.4 % and 2.5

%, respectively.

## 1. Introduction

The hygroscopic properties of atmospheric aerosol particles are vital for a proper description of their

direct and indirect effect on the radiative budget of the Earth's atmosphere (Hänel, 1976; Rader and
McMurry, 1986; Pöschl et al., 2005; McFiggans et al., 2006; Andreae and Rosenfeld, 2008;
Swietlicki et al., 2008; Cheng et al., 2008; Zieger et al., 2013; Rastak et al., 2014; and references
therein). The response of aerosol particles to changes in relative humidity (RH) can be obtained by
determining the growth factor   of aerosol particles under enhanced RH conditions. The latter is

possible by means of a hygroscopicity tandem differential mobility analyzer (H-TDMA).   The
principles of TDMA experiments have been described four decades ago (Liu et al., 1978; Rader and
McMurry, 1986), and a wide range of applications and modifications of this technique have been
reported since then (e.g. Brechtel and Kreidenweis, 2000; Weingartner et al., 2002; Mikhailov et al.,
2004; Biskos et al., 2006; Johnson et al, 2008; Nilsson et al, 2009;  Duplissy et al., 2009; Lopez-

Yglesias et al., 2014). Due to technical limitation, most of the traditional HTDMA studies have been
however conducted at RHs < 95 %. At higher humidity the HTDMA setup become less reliable due
to small variations in temperature in the second differential mobility analyzer (DMA2) result in large
uncertainty in RH (Swietlicki et al., 2008; Duplissy et al., 2009; Massling et al., 2011; Lopez-
Yglesias et al., 2014). Controlling RH accurately inside the second DMA is also challenging. At RH

= 95 % the most accurate chilled-mirror point hygrometer at typical accuracy of dew point
temperature of ~ 0.1 ℃ leads to an uncertainty in RH of ~ 0.6 %.  Uncertainties in HTDMA-derived
growth factors  can arise also from size bin width in the DMAs transfer function and from the offset
in dry particle sizing between two DMAs (Swietlicki et al., 2008; Massling et al., 2011; Suda and
Petters, 2013). Irregular structure of the initial dry particles leading to discrepancy between the

mobility equivalent and mass equivalent particles diameters is an additional source of growth factor
uncertainty (Mikhailov et al., 2004; Kreidenweis et al., 2005).  The role of these sources of
uncertainties increases significantly at RH > 90 %.

     The desire to expand the upper RH bound in HTDMA experiments   is mainly due bridge the
gap between sub- and supersaturation conditions to provide data on aerosol-water interaction through

the full range of relevant atmospheric saturation ratio (Kreidenweis  et al., 2005). Of particular
interest  are non-ideal behavior of aerosol aqueous solutions at RH approaching to 100%, water
activity-dependent  hygroscopicity (Petters et al., 2009; Wex et al, 2009; Mikhailov et al, 2013, and
size dependent partitioning effect between particle surface and volume (Ruehl et al., 2010).
Understanding these phenomena and their quantification in the near-saturated air is relevant to





aerosol-radiation and aerosol-cloud interactions (Pöschl, 2005; Andreae and Rosenfeld, 2008; Pajunoja et al., 2015).

Two high humidity tandem differential mobility analyzers (HHTDMA) with an increased upper limit in RH are described in literature. The setup described by Hennig et al. (2005) allows growth factor measurement up to 98 % RH. In their setup, the second DMA was submerged in a

temperature-controlled water bath. As a result, the temperature gradient inside the column was smaller than ± 0.1 ℃. However, due to RH was measured by dew point mirror (±0.1 ℃) the resulting growth factor error at high humidity was significant. Thus, at RH = 97.7 % the precision quoted by authors in absolute units is ±1.2 % and particle growth factor relative uncertainty is 16.6 % (±0.46 at growth factor value of 2.79). For 100 nm dry ammonium sulfate aerosol, these uncertainties result in

±121 % relative error in the retrieved hygroscopicity parameter (Suda and Petters, 2013).

The second HHTDMA setup was described by Suda and Petters (2013). This instrument allows growth factor measurement up to 99% RH. In their setup, the first DMA was neither insulated nor temperature controlled. The second DMA was thermally insulated. The temperature gradient in DMA2 was estimated from column exterior temperatures and did not exceed ±0.02 ℃.

At RH > 90 % they used calibration scans with ammonium sulfate to convert measured growth factors into RH using Extended Aerosol Inorganic model (E-AIM) (Clegg et al. 1998; Wexler and Clegg 2002). In this case, the precision in RH was ~1 % at RH near 90 % and ~0.1 % at RH of about 99 %. The averaged in the 80–99 % RH range the relative growth factor uncertainty was 2.2 % (Suda and Petters, 2013, obtained from Fig.2). The quoted uncertainty in measured hygroscopicity and

activity coefficients is ±20 %. Suda and  Petters (2013) analyzed in detail the HHTDMA- based sources of uncertainty in the thermodynamic coefficients of organic aqueous solutions. They concluded that the size-dependent bin width of the DMA transfer function, the sizing offset between two DMAs, irregularities in the dry particles (shape factor) and controlled RH are the main factors responsible for the resulting uncertainty in the growth factor and thermodynamic coefficients, as a

consequence.

Here we introduce a new HHTDMA instrument designed to overpass problems listed above such that the precision in growth factors in the 2–99.6 % RH range improved to ~0.6 % providing uncertainty in hygroscopicity and activity coefficients less than 4 %. We demonstrate these uncertainties for glucose aerosol particles above and below water saturation.


## 2.    Design of HHTDMA setup

Operating the HHTDMA at RHs above 99 % requires special operating procedures and temperature/humidity control systems. In this section, we describe the design of the various subsystems.

**2.1    HHTDMA setup and operation modes**





Figure 1 shows a sketch of the HHTDMA setup. Similar to conventional HTDMA system (Swietlicki et al., 2008; Duplissy et al., 2009) our setup consists of two DMAs (TSI 3081 type) connected in series with a humidity conditioning section between them. Both DMAs are housed in aluminum boxes and thermally insulated with 20 mm polyethylene foam (Fig.S1.1). The temperature inside each aluminum box is actively controlled using circulation thermostat (Lab. Companion, CW-05G) and two aluminum liquid heat exchangers (HRA120DR) with integrated fans. The DMA1 and DMA2 operated at 26 ºC and 25 ºC, respectively. Two Pt100 needle sensors (uncertainty ±0.015 ºC,) placed in the sheath and excess airlines in DMA2 (T4, T5, Fig.1). The temperature difference between sheath and excess flow was small enough i.e. within Pt100 sensors uncertainty. The DM1 and DMA2 were operated with a closed loop sheath air setup. The sheath and aerosol flow rates in both DMAs were 3.0 and 0.3 l min$^{-1}$, respectively.

The particle size distributions measured with the scanning mobility particle sizer (DMA 2, SMPS TSI 3080, CPC TSI 3772, TSI AIM version 9.0.0.0, Nov. 11, 2010) were fitted with a log-normal distribution function (Origin 9 software), and the modal diameter ($D_b$) of the fit function were used for further data analysis.

In our setup, the RHs of the sheath and aerosol flow are separately adjusted. It is therefore possible to use three different modes of operation (Mikhailov et al., 2004; 2009):

1. "Hydration &dehydration" ($h\&d$) mode (Mikhailov et al., 2004) or restructuring mode (Gysel et al., 2004) (Fig.1, red rectangle). This HHTDMA mode provides information about structural changes as a function of the relative humidity experienced during a cycle of humidification and drying (variable X = RH2; RH3, RH4 and RH5 <3 %). Here and below X represents the independent variable, i.e., the RH value taken for plotting and further analysis. The minimum mobility diameter observed in $h\&d$ mode ($D_{h\&d,min}$) can be used to approximate the actual mass equivalent diameter of dry particles ($D_{m,s}$), which is a prerequisite for accurate Köhler model calculations.

2. "Hydration" HHTDMA mode provides information about deliquescence phase transitions of dry particles and the hygroscopic growth of deliquesced particles as a function of relative humidity (variable X= RH3 ≈ RH4 ≈ RH5; Fig.1).

3. "Dehydration" HTDMA mode provides information about the efflorescence transition of deliquesced particles and the hysteresis loop between deliquescence and efflorescence transitions as a function of relative humidity upon particle sizing after conditioning and deliquescence at high RH. The water filled pre-humidifier is set to a high RH (>96 %) (variable X = RH3 ≈ RH4 ≈ RH5; Fig.1).





The mobility equivalent particle growth factor, $g_b$ was calculated as the ratio of the mobility
equivalent diameter, $D_b$ measured after conditioning (hydration, dehydration) to the minimum mobility
diameter $D_{b,h\&d,min}$ observed in $h\&d$ mode: $g_b = D_b/D_{b,h\&d,min}$ .

## 2.2 Aerosol generation

Aerosols were generated by nebulization ammonium sulfate (99.9 % pure, ChemCruz) or D-glucose
(99.55% pure, Fisher) aqueous solution at ~0.01 % and ~0.1 % mass concentration, respectively.
Two separate atomizers operated with particle-free pressurized air (2.5 bar, 2 l min$^{-1}$) were used. In the
regular aerosol drying mode the generated solution droplets were first dried to a relative humidity of ~3
% in the Nafion MD-700 (L = 60 cm), and then in the silica gel diffusion dryer (SDD, L = 100 cm ID
= 2 cm, r.t. = 62.8 s). The MD-700 dryer operated at a purge air flow of 5 l min$^{-1}$ with input RH below
0.3 %. The residual relative humidity at the exit of the SDD was <2% RH and close to that for sheath
flow in DMA1 (RH1, Fig. 1).   The dry aerosol (0.3 l min$^{-1}$) was passed through a bipolar
charger/radioactive neutralizer (Kr85) to establish charge equilibrium, and a differential mobility
analyzer (DMA1) to select monodisperse particles. The used two-stage drying system (pre-dryer +
SDD) provides the same humidity profile inside SDD throughout the HHTDMA experiment, which
minimizes the effect of the drying conditions on the particle morphology  and on the particle sizing as
a consequence (Mikhailov et al., 2004; 2009; Wang et al., 2010).

## 2.3    Aerosol conditioning

The Nafion conditioning tube with inner diameter of 2.2 mm used for aerosol humidification in all
HHTDMA operation modes. The length of H1, H2 and H3 Nafion conditioners is equal to 60, 120, and
240 cm, respectively (Fig.1).  In case of the H1 and the H2 exchangers, a 1 l m$^{-1}$ humidified air flow
passed through the outer tube to adjust the RH between 3% and 97 %. The outer shell of H2 Nafion tube
was filled with pure water and thus set a RH greater than 96 %. For efflorescence HHTDMA mode the
H1 and the H2 Nafion tubes were used in series (Fig.1).  In the h&d HTTDMA mode aerosol in series
flowed through a Nafion conditioner (H1, r.t. = 0.5 s), Nafion MD-700 dryer (L= 60 cm, r.t. = 27.2 s)
and SDD (r.t. = 16 s) in which the RH of the aerosol was reduced to below 2 % (Fig.1, aerosol pre-
conditioning section, red rectangle).  The residence time between aerosol pre-conditioning system and
DMA2 depends on the humidification mode; its minimum value is 6.5 s, which corresponds to r.t. in
hydration operation mode (Fig.1). This is sufficient time to achieve an equal size at given RH, provided
that there are no kinetic limitations to water uptake (Chuang, 2003; Mikhailov et al., 2004).

A 6mm (ID) Gore-Tex membrane 2-m and 0.5-m long accordingly used for sheath air (RH4; DMA
2) and aerosol flow (H3) humidification. The regulation of the humidity has been achieved by mixing





saturated and dry air in variable proportions (Humidifier, Fig.1). The saturated air was obtained by

bubbling air directly through water (not shown in Fig.1).

## 2.4 RH control

Relative humidity at several points throughout the apparatus was controlled by capacitive

sensors (RH1-RH5, Fig.1), supplemented with temperature (±0.2 °C) and atmospheric pressure sensors

(±2.5 mbar). In addition, the RH inside of the DMA2 was determined by combining sheath air

temperature and dew point temperature measured in the excess of airline (Fig.1). The accuracy of the

dew point temperature is ±0.1 °C, which in particular at 98 % RH leads an uncertainty of ± 0.6 % RH.

All RH sensors and dew point mirror were periodically calibrated using the LI-610 dew point generator

(LI-COR, USA). At RH > 90 % due to instrumental limitations the RH measurement accuracy by

capacitive RH probes and a dew point sensor drops noticeably. To circumvent this problem we used

ammonium sulfate particles as a calibration standard. Based on the Extended Aerosol Inorganic Model

(E-AIM, model II) (Clegg et al. 1998; Wexler and Clegg 2002), we converted the measured growth

factors into RH ($g_{b,E-AIM}$) (Rose et al., 2008; Suda and Petters, 2013; Rovelli et al., 2016). The

uncertainty of water activity calculations with the E-AIM for aqueous solutions of ammonium sulfate

above deliquescence relative humidity (DRH) (Clegg and Wexler, 2007) is better than $10^{-4}$, and is

negligible relative to uncertainties of the growth factor measurement. Figure 2 shows the measurement

uncertainty in RH by various methods. One can see that over a range of humidity levels the sensitivity

of the methods is noticeably different. Therefore, to minimize uncertainty in the RH determination we

used a dew point probe and capacitive sensors (RH4, RH5, Fig. 1) in the RH range of 5–80 %, and the

HHTDMA-derived ammonium sulfate growth factors at RH above 80 %. Note that at RH below 80 %

the E-AIM model parameters are based on the fit of electrodynamic balance (EDB) measurements, for

which the accuracy in relative humidity is ~1 % RH and in mass fraction of solute is about of ~1 %

(Chan et al., 1992; Clegg et al., 1995). For these uncertainties, the propagated growth factor error to

be ~1.0 %, which exceeds instrumental growth factor error by a factor of ~5 (see next section).

Consequently, below deliquescence transition the RH accuracy was calculated accounting for EDB-

based growth factor error (Fig.2, upper black curve), whereas above the deliquescence transition RH

accuracy were obtained using instrumental growth factor uncertainty (Fig. 2, lower black curve).

## 2.5 Growth factor uncertainty

The instrumental growth factor error depends on uncertainty in particle sizing, which is result of

variations in the flow rate, voltage, temperature, and atmospheric pressure. Uncontrolled change in these

parameters causes a drift in dry mobility diameter and measured growth factor as a consequence.



Regarding the precision of particle sizing by DMA1, the voltage variation from the specified value is less than $\pm 1 \cdot 10^{-4}$ (HCE 7 -12 500, FuG Electronic), the relative standard deviation of the sheath flow is 0.06 %. Unlike DMA1, where the critical orifice maintains a constant sheath flow, in the DMA2 the sheath flow is monitored by the microprocessor using temperature and pressure sensors built into mass flow meter. Our test measurements showed that within 10 hours, which is typical time scale of HHTDMA measurements, no trend in dry mobility diameter was observed (Fig.S.1.2). Over entire period statistic error of the selected dry mobility diameter of 99.31 nm to be 0.16 нм ($2 \times \sigma_s$) which is propagated in the instrumental relative growth factor error of $\pm 0.002$ ($\sigma_s$ is the standard deviation of selected dry mobility diameter). Nevertheless, to minimize systematic error caused by the casual drift of initial dry mobility diameter its size was measured at the beginning and at the end of every experiment.

We checked effect of width of the DMA2 transfer function on the uncertainty in particles sizing by measuring the variability of the selected dry particles with diameters of 100, 200 and 300 nm. For these diameters based on six repeated measurements the relative uncertainty $2\sigma_s/D$ was 0.0016; 0.0022 and 0.0015, respectively, indicating that the effect of transfer function broadening on the particle growth factor is negligibly small. However, variation in RH within DMA2 significantly affects the measurement precision of particle diameters, especially at high humidity. The RH-dependent measurement uncertainty in $D_{b,RH}$ was fitted by the 3-parameter exponential function (Fig.S1.3):

$$\frac{2\sigma_{b,RH}}{D_{b,RH}} = \alpha + \beta \cdot exp(\varepsilon \cdot RH) \tag{1}$$

Here $\sigma_{b,RH}$ and $D_{b,RH}$ are the standard deviation and particle mobility equivalent diameter at a given RH. The fit parameters ($\alpha, \beta$ and $\varepsilon$) obtained for ammonium sulfate and glucose aerosol particles are listed in supplement (S1). Finally, HHTDMA-derived growth factor uncertainty was calculated as follows:

$$\Delta g_b = \left( \left[ \left(\frac{2\sigma_s}{D_{b.s}}\right)^2 + \left(\frac{2\sigma_{RH}}{D_{b.RH}}\right)^2 \right] g_b^2 + \left(\Delta RH \frac{dg_b}{dRH}\right)^2 \right)^{1/2}, \tag{2}$$

where the terms in square brackets describe the instrumental uncertainty of $g_b$, and the next term accounts for the contribution of the RH sensor uncertainty to the particle growth factor. Note, when using Eq. (2) the $dg_b/dRH$ was substituted by the measured $\Delta g_b/\Delta RH$.

We also checked the sensitivity of the SMPS inversion algorithm and log-normal fit to the ammonium sulfate particle size variations exiting DMA1. Figure 3a shows response of the SMPS classifier to a voltage (size) change in DMA1. It is seen that 1-volt step causes a proportional displacement of the particle diameter by 0.12 nm (linear fit). Inset in Fig.3a indicates that this resolution significantly exceeds the size of individual bin (shown bin midpoints). As an example Fig.3b shows SMPS histogram of number particle distribution obtained for two DMA1 selected particles with $\Delta = 3.9$

volt. It can be seen that voltage shift causes a change in particle concentration in each size bin, leading
to a corresponding shift of the fitted size distribution and a change in modal particle diameter by 0.5 nm

as a result (insert in Fig.3b). Thus, the growth factor of near-monodisperse particles can be determined
with higher precision than resolution of the size bins in the SMPS-derived histogram.

To eliminate the uncertainty in growth factors arising from the sizing offset between two DMAs
(Massling et al., 2011) in our instrument the dry mobility diameter selected by DMA1was measured by
the DMA2 on a par with the wet mobility diameter.  However, additional uncertainty is introduced dew

to particle shape factor. As will be shown below, we managed to minimize this uncertainty using the
restructuring mode.

## 3    Aerosol particles shape

Inorganic and organic aerosol particles as well as their mixtures restructure upon humidification below

its deliquescence (Mikhailov et al., 2004, 2009; Biskos et al., 2006; Gysel et al. 2004).   Irregular
envelope shape and porous structure can cause a discrepancy between the mobility equivalent and mass
equivalent particle diameters. To account for restructuring we use the minimum mobility particle
diameter, $D_{b.h\&d,min}$     obtained in $h\&d$ HHTDMA mode as an approximation of   mass equivalent
diameter of the dry solute particle, $D_{m,s}$ i.e. $D_{m,s} = D_{b,h\&d,min.}$. Based on $h\&d$ HHTDMA measurements

the dynamic shape factor, $\chi$ of the dry initial particles can be estimated as following (DeCarlo et al.,
2004):

$$\chi = \frac{D_{b,i}C\left(D_{b,h\&d,min}\right)}{D_{b,h\&d,min}C\left(D_{b,i}\right)} \tag{3}$$

where $D_{b,i}$  is the initial mobility equivalent diameter selected by DMA1 and measured by DMA2,
$C(D_{b,h\&d,min})$ and   $C(D_{b,i})$ are the Cunningham slip correction factors for the respective diameters
$D_{b,h\&d,min}$ and  $D_{b,i}$ (Willeke and Baron, 1993). $\chi$ can be split into a component $\beta$ which describes the

shape of the particle envelope and a component $\delta$ which is related to the particle porosity and allows
the calculation of the void fraction inside the particle envelope $f$ (Brockmann and Rader, 1990):

$$\chi = \beta\delta\frac{C\left(D_{h\&d,min}\right)}{C\left(D_{h\&d,min}\delta\right)} \tag{4}$$

$$f = (1 - \delta^{-3}) \tag{5}$$

## 4    Thermodynamic models

### 4.1    Full Köhler model





In this study, we used full Köhler model (Brechtel and Kreidenwies, 2000; Rose et al., 2008; Mikhailov et al., 2009) as a basis for HHTDMA calibration and for comparison to the measured growth factor-RH dependences:

$$\frac{RH}{100} = a_w \, exp\left(\frac{4\sigma \bar{V}_w}{RTD_m}\right),$$   (6)

where $a_w$ is the water activity, $\sigma$ is the surface tension of the solution droplet, $\bar{V}_w$ is the partial molar volume of water in solution, $R$ is the ideal gas constant, $T$ is the droplet temperature and $D_m$ is the mass

equivalent droplet diameter.

The partial molar volume of water in the droplet solution can be expressed by (Brechtel and Kreidenweis, 2000)

$$\bar{V}_w = \frac{M_w}{\rho}\left(1 + \frac{X_s}{\rho}\frac{d\rho}{dX_s}\right),$$   (7)

where $M_w$ is the molecular weight of water, $\rho$ is the density of the solution, and $X_s$ is the mass fraction of solute in the droplet.

The ratio of the aqueous droplet diameter, $D_m$, to the mass equivalent diameter of a particle consisting of the dry solute, $D_{m.s,}$ is defined as the mass equivalent growth factor, $g_m$:

$$g_m = \frac{D_m}{D_{m.s}} = \left(\frac{\rho_s}{X_s\rho}\right)^{1/3}.$$   (8)

The concentration dependence of $\rho$ for ammonium sulfate aqueous solution can be taken elsewhere (Tang and Munkelwitz, 1994). The density for glucose solution was obtained by the 2-nd order polynomial fit of the experimental data reported by Cerdeirina et al. (1997) ($X_s$ <0.5) and Taylor and

Rowlinson (1955) ($X_s$ < 0.8):

$$\rho(g\,cm^{-3}) = 1.0008 + 0.3477X_s + 0.1692X_s^2$$   (9)

with standard deviation of the fit is 0.0021 g cm$^{-3}$. The surface tension of the aqueous solution can be obtained using a simple linear approximation:

$$\sigma = \sigma_w + \sigma_{conc}[Concentration],$$   (10)

where $\sigma_w$ = 72.0 mN m$^{-1}$ is the surface tension of pure water at 25ºC and $\sigma_{conc.}$ account for the influence of the droplet composition and units of concentration. For ammonium sulfate $\sigma_{conc.}$ = 2.17 mN kg mol$^{-1}$

(molality-based) (Hänel, 1976) and for glucose solution $\sigma_{conc.}$ = 0.29 mN l mol$^{-1}$ (molarity-based) (Aumann et al., 2010). Solute molality, $\mu_s$ (mol kg$^{-1}$), solute molarity, $C_s$ (mol l$^{-1}$), molecular weight of solid, $M_s$ (g mol$^{-1}$), solution density, $\rho$ (g cm$^{-3}$), mole fraction of solute, $x_s$ , mole fraction of water, $x_w$ ($x_w$ =1- $x_s$), and mass fraction of solute, $X_s$ are related by:





$$C_s = \frac{\mu_s \rho}{1 + \mu_s M_s / 1000} \ , \tag{11}$$

$$X_s = \left(1 + \frac{1000}{\mu_s M_s}\right)^{-1}, \tag{12}$$

$$x_s = \frac{X_s / M_s}{X_s / M_s + (1 - X_s)/M_w}. \tag{13}$$

$$\mu_s = \frac{x_s \cdot 1000}{(1 - x_s) \cdot M_w} \ . \tag{14}$$

In the full Köhler model calculations $a_w$ of the ammonium sulfate particles was taken from the Extended
Aerosol Inorganics Model (E-AIM, model II) (Clegg et al. 1998; Wexler and Clegg, 2002) and the
corresponding molality $\mu_s$ was obtained. Alternatively, water activity of the glucose solution droplets
was obtained from relation:

$$a_w = \gamma_w x_w , \tag{15}$$

where water activity coefficient, $\gamma_w$ calculated from two-suffix Margules equation (Taylor and
Rowlinson, 1955):

$$ln\,\gamma_w = -A x_s^2, \tag{16}$$

with $A$ = -1.957 (± 0.062). Note, there are also other theoretical equations such as three-suffix Margules
equation (Cindio and Correra, 1995; Miyawaki et al., 1997), but the difference in the $\gamma_w$ calculated values
between Eq. (16) and more complicate expressions is negligibly small within ~0.01%. Equations (6–
16) can be used to model the hygroscopic growth of aerosol particles, i.e., to calculate $g_m$ and $D_m$,
respectively, as a function of $D_{m.s}$ and $RH$.

## 4.2    Growth factor and hygroscopicity parameterization

As proposed by Kreidenweis et al. (2005), hygroscopic growth data points can be approximated with a
polynomial 3-paramter fit function of the following form:

$$g_b = \left(1 + [k_1 + k_2 a_w + k_3 a_w^2]\frac{(1 - a_w)}{a_w}\right)^{1/3}. \tag{17}$$

Using Eq.(6) we convert the measured RH-based growth curves ($g_b$ vs. RH) into water activity growth
curves ($g_b$ vs. $a_w$) assumed that $D_{b,h\&d.min} = D_{m,s}$ , $\bar{V}_w$  and $\sigma$ equal to the partial molar volume and
surface tension of pure water, respectively. For pure glucose aerosol particles, the relative errors
introduced by this simplifying assumption in the calculation of $a_w$ from Eq. (6) were less than 0.1%.





According to Petters and Kreidenweis (2007), the hygroscopic properties of aerosol particles can be
approximately described by a single hygroscopicity parameter, $\kappa$ :

$$a_w = \frac{D_m^3 - D_{m.s}^3}{D_m^3 - D_{m.s}^3(1 - \kappa)} , \qquad (18)$$

Under the assumption of volume additivity, Eq. (18) can be rewritten as:

$$\kappa = \frac{(g_m^3 - 1)(1 - a_w)}{a_w} . \qquad (19)$$

As a result, the hygroscopicity $\kappa$ can be determined from each HHTDMA-measured data pairs of $g_m$ vs.
$a_w$ under the assumption $g_b = g_m$. For ideal solution, Raoult $\kappa$, $\kappa_R$ can be calculated using known
constants (Rose et al., 2008; Mikhailov et al. 2009);

$$\kappa_R = v_s \frac{M_w \rho_s}{M_s \rho_w} , \qquad (20)$$

where $v_s$ is the stoichiometric dissociation number of solute.

### 4.3 Molal osmotic coefficient

According to Robinson and Stokes (1970) the molal osmotic coefficient of solute in aqueous solution,
$\Phi_s$ can be obtained from relation:

$$\Phi_s = -\frac{1000 ln a_w}{v_s \mu_s M_w} . \qquad (21)$$

For hydrophilic nonelectrolytes ($v_s$ =1) nonideality is caused by hydration of solutes. As proposed by
Rudakov and Sergievski (2009) for such aqueous solutions the activity coefficient of water can be
estimated according to the equation:

$$ln\, \gamma_w = 2h^0(ln x_w + x_s) , \qquad (22)$$

where $h^0$ is the hydration number of the solute at $x_w = 1$. From Eq.(15) and Eq.(22) it follows:

$$\Phi_s = -\frac{x_w}{1 - x_w}(ln x_w + 2h^0[ln x_w + (1 - x_w)]) . \qquad (23)$$

### 4.4 HHTDMA-derived activity coefficients

In a binary system at constant temperature and pressure, the activity coefficient of water, $\gamma_w$ and activity
coefficient of solute, $\gamma_s$ are related by the Gibbs–Duhem equation:

$$x_w dln\gamma_w + x_s dln\gamma_s = 0 . \qquad (24)$$

$x_w$ can be obtained based on HHTDMA-derived aerosol particle growth factors. First simple method is
based on volume additivity assumption when the volume of the solution droplet is given by the sum of





the volumes of the dry solute and of the pure water contained in the droplet (Mikhailov et al., 2009;
Petters et al., 2009):

$$\frac{1}{x_w} = 1 + \frac{\rho_s M_w}{\rho_w M_s}(g^3 - 1)^{-1}. \tag{25}$$

If the concentration dependence of the solution density is known, then $x_w$ can be obtained without
assumption of volume additivity by iteratively solving Eq. (8) with other equation where $\rho$ and
concentration is given explicitly. For example, Eq. (9) was used for glucose solution droplets. The mass
fraction, $X_s$ calculated in this way for a given $g_m$ was then converted into $x_w$, using Eq.(13).

The activity coefficient, $\gamma_s$ of glucose in water solution was obtained by numerical integration of
Eq.(24) using EXPGro3 function (Origin 9 software) to fit and then integrate experimental dependence
of $x_w/x_s$ vs. $\ln\gamma_w$. The boundary conditions are based on asymmetric reference system: at $x_s \to 0$; $\gamma_w \to$
1, $\gamma_s \to 1$, i.e., at $x_s = 0$; $\ln\gamma_w = 0$ and $\ln\gamma_s = 0$. Integration yields:

$$\ln\gamma_s(at\ x_s) = -[F(\ln\gamma_w\ at\ x_s = x_s) - F(\ln\gamma_w = 0\ at\ x_s = 0)] \tag{26}$$

Using Eq. (15) the received $\gamma_s$ can be easily converted into the solid activity, $a_s$. Thus, relying only on
known solution density, the thermodynamic parameters $x_w$, $\gamma_w$, and $a_w$ as well as $x_s$, $\gamma_s$, $a_s$ and $\Phi_s$ can
be obtained from the HHTDMA-measured $g_b(RH)$ dependence without assumption of volume
additivity. This is important for concentrated droplet solution where volume additivity is not always
hold.

**4.5 Surface adsorption**

The amount of water adsorbed on the surface of crystalline aerosol particles prior to deliquescence
can be described with surface coverage ($\Theta$) (or number monolayers on dry particles surface). Assuming
that initial particles are compacted and spherical, the number monolayers can be calculated from the
ratio:

$$\Theta = \frac{D_{RH} - D_{m,s}}{2D_w}, \tag{27}$$

where $D_w$ is the diameter of adsorbed water molecule (0.277 nm) (Yeşilbaş et al., 2016).
The FHH (Frenkel, Halsey and Hiil) isotherm is the frequently used equation that relates surface
coverage to a water activity:

$$a_w = exp(-A_{FHH}/\Theta^{B_{FHH}}), \tag{28}$$





where parameter $A_{FHH}$ characterizes interactions between adsorbed molecules in the first monolayer and between the surface, and $B_{FHH}$ characterizes the attraction between the solid surface and the

adsorbate in subsequent layers. Inserting Eq.(28) into Köhler model (Eq. 6) the parameters $A_{FHH}$ and $B_{FHH}$ can be estimated (Romakkaniemi et al., 2001; Sorjamaa and Laaksonen, 2007; Hatch, et al., 2019).

## 5    Experimental results and discussion

### 5.1. Aerosol particle restructuring

The hydration&dehydration (h&d) HHTDMA operation mode was first used to study effect of the drying condition on the aerosol particle restructuring. Specifically, in the aerosol generation section (Fig.1) we alternatively used the Nafion MD -700 dryers of various lengths providing residence time of the aerosol flow in the range of 27– 62 s, solely the silica gel diffusion dryer (SDD) with r. t. = 62 s, and a coupled drying system, comprising the Nafion MD -700 and the SDD dryers. The dried aerosol

particles selected by DMA1 entered to the pre-conditioning section (Fig.1, red rectangle), where during a cycle of humidification (H1, r.t. = 0.5 s) and drying (Nafion MD 700, r.t. = 27 s; SDD, r.t. = 16 s) they underwent microstructural transformation, as previously described in Mikhailov et al. (2004; 2009).

#### 5.1.1    Ammonium sulfate particle

Figure 4a shows the change in the initial dry mobility diameter of 100.3 nm ammonium sulfate aerosol particles obtained at different drying conditions. In the range of 2–60 RH the mobility diameter gradually decreases, and when RH is more than 70 % RH it becomes almost constant with $D_{b,h\&d,min}$ observed at 80–90% RH. The $D_{b,h\&d,min}$ values obtained for all drying modes are shown in Table 1.

       Interestingly, when using only MD-700 dryer the variation in the r.t. from 27 to 67 seconds leads

to a decrease in the $D_{b,h\&d,min}$ by only 0.4 nm (i.e. at r.t. =27 s, $D_{b,h\&d,min}$ = 98.4 nm; at r.t. = 67s, $D_{b,h\&d,min}$ = 98.0 nm). A sharp decrease in minimum mobility diameter by ~ 3 nm ($D_{b,h\&d,min}$ = 95.3 nm) occurs when SDD (r.t. = 62 s) is added to the MD-700 membrane dryer (r.t. = 27 s), that is already effloresced aerosol particles (RH ~ 3% at the outlet of the MD-700 dryer) undergo further microstructural changes inside SDD. The maximum reduction (by ~7 nm) of the initial DMA1 selected particles was observed

when only SDD was used as a desiccant, at which $D_{b,h\&d,min}$ = 93.2 nm and 93.5 nm (first and second run with the same SDD, Fig.3a).

       Multiple Köhler model calculations based on $D_{b,h\&d,min}$ obtained in all used drying modes are in excellent agreement with the observed hygroscopic growth curves. These findings confirm the compactness and spherical shape of dry particles, despite the fact that the absolute values of $D_{b,h\&d,min}$

are different and strongly depend on the drying conditions (Fig.4a).



The different values of $D_{b,h\&d,min}$ observed upon $h\&d$ mode is a result of different microstructure of the initial dry particles having the same $D_{b,i}$ (Table 1). As previously noted the dry particle morphology depends on drying rate (Zelenyuk et al. 2006, Zhao et al., 2008; Mikhailov et al., 2009; Wang et al, 2010), since the solidification of aerosol droplets is mainly governed by kinetic rather than thermodynamic factors. Experiments with MD -700 membrane dryer show that at the same drying rate, the residence time has a little effect on ammonium sulfate particle morphology, i.e. an increase in the r.t. from 27 to 67 s., the dynamic shape factor slightly grows from 1.033 to 1.040 (Table 1). The obtained in this experiment values are close to those reported by Kuwata and Kondo (2009), who used the combination of a DMA and an APM (aerosol particle mass analyzer) system. They estimated that $\chi$ of $(NH_4)_2SO_4$ for 50-150 nm is 1.01 ~ 1.04. Zelenyuk et al. (2006) using DMA and single-particle laser ablation time-of-flight mass spectrometer (SPLAT) measurements showed that $\chi$ is $1.03 \pm 0.01$ at 160 nm. Biskos et al. (2006) estimated a $\chi$ for 6–60 nm $(NH_4)_2SO_4$ particles of 1.02, based on the observed particles restructuring in the hydration HTDMA mode. Our $\chi$ values obtained with MD-700 membrane dryers and those that preliminary reported most likely reflects surface irregularities as observed by electron microscope (Dick et al, 1998; Zelenyuk et al., 2006). At the same time, an experiment with serially connected dryers (MD -700 + SDD) indicates that after first dryer effloresced particles still contain liquid. The liquid could be located in cavities with various degrees of shielding (Cohen et al., 1987; Weis and Ewing, 1999; Colberg et al., 2004). Figure 5 outlines possible structures of aerosol particles and their microstructural rearrangements during $h\&d$ experiment. Based on the experimental results considered above, we can assume that in membrane dryers (MD-700) the effloresced particles release surface water and water that is stored in relatively open cavities, providing an irregular aerosol particles envelope (Dick et al., 1998). However, part of liquid remains in either completely closed or partially shielded cavities. The latter can be pores, veins, and grain boundaries, that retain water due to inverse Kelvin effect on concave surfaces (Fig. 5, pattern I) thereby impeding exchange between gas phase and the cavities. The fresh SDD added to membrane dryer overcome diffusion barrier created by capillary forces most likely due to drying rate in SDD is much faster than that in membrane dryer. As a result, some of partially shielded cavities release excess water. Note this process is accompanied by further microstructural rearrangement, leading to formation porous aerosol particles (Fig.5, pattern II) with $D_{b,h\&d,min}$ is smaller than that observed after membrane dryer (~ 95.3 nm vs. ~ 98.4 nm, Table 1). Accordingly, the aerosol particle shape factor, $\chi$ increases from ~ 1.03 to ~ 1.09 (Table 1). Finally, more porous particles are formed when solely fresh SDD is used as a desiccant ($D_{b,h\&d,min}$ = 93.3 nm, $\chi$ = 1.13; Table 1). As pointed out before (Mikhailov et al., 2009; Wang et al., 2010) excess charge and rate of drying are important for microstructure of aerosol particles generated by nebulization of aqueous solution. Most likely in case of fresh SDD the most porous/irregular particles (Fig.5, pattern III) forms due to strong kinetic limitations arising at a sufficiently rapid drying inside SDD. More pronounced





multiple nucleation events could occur by increasing the number of polycrystals and accordingly the number and scale of cavities. As a result, at the same $D_{b,i}$ the observed $D_{b,h\&d,min}$ of re-dried particles (h&d mode) is strongly dependent on the drying conditions (Table 1; Fig.5, the last column).

430        Assuming that the $\chi$ value obtained in the experiment with the membrane dryer accounts for the aerosol particles envelope, i.e. $\chi = \beta = 1.033$ and using Eq.(4) and Eq.(5) we estimated the particle porosity, $\delta$ and void fraction, $f$, respectively. The calculated values of the particle void fraction for the coupled (MD 700 + SDD) drying system and for the single SDD are ~ 9% and ~ 14%, respectively (Table 1). Obviously, this difference reflects the effect of drying conditions on particle morphology as discussed above. Since in the two-stage drying system, the microstructural rearrangement of particles

inside SDD occurs due to the remaining water the obtained $f \approx 9$ % can be attributed to the volume fraction of water stored in pores and veins after first drying stage (Fig.5 pattern I). In the mole fraction basis the water content in the dry solid aerosols can be obtained from:

$$\frac{n_w}{n_s} = (\delta^3 - 1)\frac{\rho_w M_s}{\rho_s M_w}. \tag{29}$$

According to Eq. (29) the $H_2O:(NH_4)_2SO_4$ molar ratio, $(n_w/n_s)$ in the dry particles after membrane dryer is ~ 0.4 (0.41 and 0.39 for the first and second run). In case of the one-stage of aerosol drying

when only SDD is used the volume fraction of ~14 % corresponds to $H_2O:(NH_4)_2SO_4$ molar ratio of ~0.7 (0.69 and 0.73 for the first and second run). This value can be considered as an upper limit of the water content, since it is assumed that all cavities were filled with water. The $H_2O:(NH_4)_2SO_4$ molar ratio range of 0.4–0.7 obtained in this study at RH < 3 % is close to that reported by Weis and Ewing (1999) for submicron NaCl aerosol particles with median diameter of 350 nm. In their FTIR

spectroscopic flow tube experiment for RH of 15– 5 % the obtained $H_2O:NaCl$ molar ratio in the silica gel dried particles varies between 0.5 and 0.7. They suggest that during the crystallization process water is present in open and shielded pockets.

### 5.1.2 Glucose particles

Figure 4b shows mobility equivalent diameter observed upon h&d mode of glucose aerosol particles. Unlike ammonium sulfate (Fig.4a), the minimum mobility diameter of the glucose aerosol particles is already observed at ~ 20 % RH. Moreover, $D_{b,h\&d,min}$ was practically independent of drying conditions. Accordingly, the shape factor calculated from Eq. (3) is also constant ($\chi$ =1.06 ; Table 1). Atomic force microscopy analysis performed by Estilliry et al. (2017) indicates that 100 nm glucose particles as well

as other sugars are perfect spheres therefore, one can assume that $\beta$ =1. From Eq. (4) and Eq.(5) it follows that $\delta$ ~ 1.03 and $f$ ~ 10 % (Table 1). The fact that particle restructuring does not depend on the residence time and type of dryer indicates a lower energy barrier to the water transport from the cavities



to ambient air as compared to the ammonium sulfate particles. As will be shown below glucose aerosol particles like other monosaccharides tend to reversible water uptake starting at very low RH, which is

typical for particles with an amorphous structure (Fig.5, pattern IV) (Mikhailov et al., 2009; Koop et al., 2011). The absorbed water acts as a plasticizer, which soften the microstructural rearrangements inside swelling particles (Fig. 5). Moreover, at low RH the water uptake is facilitated by presence of alcohol functional groups within sugars that form hydrogen bonds with water. These effects can explain why in contrast to ammonium sulfate, restructuring of the glucose aerosol particles starts at low

humidity and practically completed at ~ 20 % RH. A slight increase in $D_{b,h\&d}$ observed at RH above 20 % RH  (Fig. 4b) is probably due to  high hygroscopicity of glucose and low diffusivity of water molecules through a (semi)-solid matrix of the compacted particles (Fig. 5) (Shiraiwa et al., 2013). However, the fact that in case with glucose aerosol particles the drying conditions do not have a significant effect on $D_{b,h\&d,min}$ is not entirely clear. Further work is needed to clarify this effect.

470        One of the most important structural metrics of a porous material is the connectivity of the pore space, or the so-called pore network. This has bearing on the diffusive tortuosity and permeability of water. Most of the pores are connected to each other as well as to the surface via small throats (open pores), whereas some pores are shielded from the connected structure.  According to estimates, the fraction of voids in the aerosol particles of ammonium sulfate and glucose obtained under the same

drying conditions are comparable (Table 1). However, pore network can be different. It is possible therefore, that in amorphous glucose the pore network has more open pores than in case of polycrystalline ammonium sulfate particles, leading to a more efficient exchange of water between filled cavities and gas phase at the same drying conditions.

**5.2    Hygroscopic growth**

To avoid the uncertainties associated with the aerosol particles morphology, we combined h&d mode with one of the hygroscopic growth mode. That is before aerosol particle humidification (hydration, dehydration) the dry aerosol particles selected by DMA1 first entered to the pre-conditioning (PC) section (Fig.1, red rectangle) where they underwent microstructural rearrangements forming more

compact and near-spherical particles. In the PC section, relative humidity (RH2, Fig.1) was maintained in the range of 80–90 % and ~ 20 % for ammonium sulfate and glucose aerosol particles, respectively. These RH values correspond to $D_{b.h\&d,min}$ obtained in h&d mode (Fig.4) which is a good approximation of  mass equivalent diameter of dry particles.

**5.2.1    Ammonium sulfate water uptake prior to deliquescence.**



Figure 6 shows the change in the initial mobility diameter of ammonium sulfate particles observed in hydration mode before deliquescence transition. It is seen that particles with and without pre-conditioning demonstrate different response to changes in RH. The particles that bypass pre-conditioning step due to irregular/porosity microstructure undergo a strong restructuring ($D_{b,i}$ decreased by 4.3 nm), while pre-conditioned particles (h&d mode) exhibit a small but continuous hygroscopic growth. Assuming that initial pre-conditioning particles are compact and spherical (i.e $D_{b,i} = D_{b,h\&d,min} = D_{m,s}$ and using Eq. (27) we have converted the difference between the mobility diameters observed in hydration mode into an equivalent number of monolayers. As illustrated in Fig. 6 we obtained near liner growth of $\Theta$ from ~ 0 to 3.5 for the range of 5–75 % RH and sharp increase up to $\Theta$ ~ 6 over the range of 75–79 % RH. The findings are consistent with our earlier HTDMA studies of water adsorption on ammonium sulfate particles (Mikhailov et al.2009) and with the literature data considered therein. The obtained $\Theta$(RH) dependence was fitted using Eq. (6) with water activity taken from FHH isotherm Eq. (28). Calculations were performed assuming that $\sigma$ and $\bar{V}_w$ parameters equal to those for pure water. The fit result is shown in Fig. 6 (solid line). The best fit parameters are $A_{FHH}$= 1.07 ± 0.08 and $B_{FHH}$= 0.94 ± 0.07. Romakkaniemi et al. (2001) also used HTDMA measurements with NaCl and $(NH_4)_2SO_4$ particle between 8 and 15 nm to estimate $\Theta$ before deliquescence transition. For ammonium sulfate particle calculated parameters for FHH isotherm were $A_{FHH}$= 0.68 and $B_{FHH}$= 0.93. The $A_{FHH}$ value obtained in our work is ~40 % higher than that reported by Romakkaniemi et al. (2001). One possible explanation is that the surface coverage is lower on surfaces of nano-sized particles compared to flat surfaces (Müller at al., 1987). Another and perhaps more important reason is that Romakkaniemi et al. (2001) used initial mobility diameter, $D_{b,i}$ for $\Theta$ calculation without particles shape correction. Biskos et al. (2006) have shown that nano-sized ammonium sulfate particles in range of 6-60 nm undergo a restructuring upon RH increasing. Thus for 6–8 nm particles the minimum mobility diameter observed in the 30–60 % RH range was by ~2 % lower than initial mobility diameter, that corresponds to the uncertainty in $\Theta$ about ~1 monolayer of adsorbed water (see Eq.27). This explains why the Romakkaniemi et al. (2001) data of $\Theta$ are lower than that obtained in this study.

The water uptake before particles deliquescence was detected in earlier studies (Weingartner et al., 2002; Gysel et al., 2002; Biskos et al., 2006), but it was observed mainly at RH >70 %. Most likely restructuring, which occurs upon particles hydration has masked water uptake at low RH. Experimental $D_b$ (RH) dependence obtained for non-pre-conditioning particles (Fig.6, open circles) clearly demonstrate this effect. Alternatively, the hygroscopic growth data obtained for pre-conditioning particles with compact structure (Fig.6, closed circles) shows that the water adsorption on the surface of the solid particle occurs already at lower humilities, ranging from ~15 % RH.





### 5.2.2 Ammonium sulfate hydration and dehydration.


Figure 7 shows growth factors of the pre-conditioned ammonium sulfate particles with $D_{b.h\&d,min} = 79.6$ nm obtained upon hydration and dehydration HHTDMA modes at 298 K. The observed efflorescence RH (ERH = 34.8 ± 0.2 %) and deliquescence RH (DRH = 79.9 ± 0.2 %) are within literature data obtained for submicron ammonium sulfate particles (Mikhailov et al., 2009; Gao et al., 2006; Ciobanu et al., 2010; and references therein). The experimental growth factors are compared to a full Köhler model with water activity parameterization derived from the E-AIM II (Clegg et al. 1998; Wexler and Clegg 2002). As illustrated in Fig.7a up to 97% RH the HHTDMA experimental data are in very good agreement with model growth factors. Insets in Fig.7a shows the experimental growth factor uncertainties which gradually go up due to increase of the RH uncertainty (Eq. 2, terms $2\sigma_{RH}/D_{b,RH}$ and $\Delta RH(dg_b/dRH)$). Averaged over the range of 38–96 % RH, the mean relative deviation between measurement and model results were < 0.5 %. The good agreement between model and measurement results confirms that the ammonium sulfate particles with $D_{b,h,min}$ were compact and spherical (i.e. $D_{b,h,min} = D_{m.s.}$). However above ~97 % RH due to sharp growth of the $\Delta RH(dg_b/dRH)$ term in Eq. (2) the observed growth factors are systematically deviate from Köhler model. Thus at ~98 % RH this deviation is ~ 7 %, and at 99.5 % RH it is already ~15 % (insert in Fig.7a).




Figure 7b shows the measured growth factors, which were converted into RH using E-AIM at RH above deliquescence transition. In this case, the RH accuracy is determined by the instrumental growth factor error (Eq. 2, terms in square brackets). Inserts in Fig.7b indicate that RH accuracy progressively improves with RH increasing. Thus at 85% RH absolute accuracy is ±0.3%, while at 99.5% RH it is only ±0.03% (Fig. 2). Thus, using experimental ammonium sulfate growth factors, it is possible to eliminate RH uncertainty generated by capacitive and dew point sensors at RH above 80%.


Overall, the combination of two HHTDMA operation modes (h&d and hydration/dehydration) that eliminate the effect of particle shape factor, and precise determination of RH using ammonium sulfate, is a prerequisite for accurate determination of the thermodynamic parameters of aerosol particles in the wide range of RH. In the next section, we will show the effectiveness of this approach by the example of glucose aerosol particles.


### 5.2.3 Glucose hydration and dehydration

Figure 8a shows mobility equivalent growth factor observed upon hydration and dehydration of pre-conditioned glucose aerosol particles with $D_{b.h\&d,min} = 99.6$ nm observed upon hydration and dehydration HHTDMA operation modes. In both modes, over the 2-99.6 % RH range the change in the growth factor occurred gradually. In contrast to (poly-) crystalline ammonium sulfate (Fig.7a), no stepwise changes in $g_b$ associated with DRH and ERH phase transitions were observed. Such behavior






is a typical for particles with amorphous structure as earlier discussed in Mikhailov et al. (2009). In

general, growth factors obtained in hydration and dehydration experiments are in a good agreement with those previously reported by Mochida and Kawamura (2004) and Suda and Petters (2013). Nevertheless, a slight positive deviation of ~ 1% can be traced throughout the all RH range. Growth factors presented by Mochida and Kawamura (2004) and Suda and Petters (2013) were calculated without particle shape correction. As noted in Sect.5.1.2 due to porosity the glucose aerosol particles undergo a wet

restructuring decreasing their initial mobility diameter (Fig.4b; Table 1)). Therefore, using $D_{b,i}$ instead of $D_{b,h\&d.min}$ as an approximation of mass equivalent diameter of the dry solute particle $D_{m,s}$ may lead to underestimated values of the growth factor (Eq.8).

Figure 8b shows the change in the total relative uncertainty of the growth factor caused by RH and instrumental uncertainties. A small drop in growth factor uncertainty observed at ~80% RH (blue

curve) is caused by replacing of the RH control method (Sect.2.4) and its decrease above ~97% RH is due to a sharp drop in ΔRH ($g_{b,E-AIM}$) near water saturation (Eq. (2); Fig.2). On the contrary, the instrumental $g_b$ uncertainty increases monotonously (Fig.8b, red curve) due to the smooth growth of the RH dependent the $\sigma_{b,RH}/D_{b,RH}$ ratio in Eq. (1) (Fig. S1.3).

Figure 9 shows the HHTDMA growth factors as compared to full Köhler model (Eq.6) with $a_w$

calculated from Eq.(15) using bulk water activity coefficient, $\gamma_w$ from Taylor and Rowlinson (1955) (Sect.4.1), and $\bar{V}_w$ and σ calculated from Eq.(7) and Eq.(10), respectively. Excellent agreement between HHTDMA-based and full Köhler model data is observed: throughout the 91.0 - 99.6% RH range average deviation of the experimental data points from the model is 0.7%. The observed coincidence indicates that $D_{b,h\&d.min}$ value obtained upon restructuring is a good approximation of mass equivalent

diameter of the dry glucose aerosol particle, i.e. $D_{b,h\&d.min} \approx D_{m.s} = 99.6$ nm. It also confirms the small growth factor uncertainty associated with RH and instrumental $g_b$ errors, which is in the range of 0.3-0.9% throughout the all 2 - 99.6% RH interval (Fig.8b, black curve).

### 5.3    Glucose thermodynamic variables

### 5.3.1    Water activity and hygroscopicity parameter

Using Eq. (6) the experimental $g_b$ vs. RH data points were converted into data pairs $g_b$ vs. $a_w$ (Sect. 4.2). Figure 10a shows the obtained activity-based growth factors, which were fitted with Eq.(17) to determine best-fit values of the parameters $k_1$, $k_2$ and $k_3$ (Table 2). Figure 10a also illustrates the difference between experimental data points and ideal solution model, which traced throughout the all

water activity range (inserts in Fig.10a) with maximal deviation of 3.6% at $a_w \approx 0.8$. Only at $a_w > 0.98$ the experimental growth factors coincide with the ideal model within uncertainty of $g_b \sim 0.6\%$. For ideal



solution model the $g_b(a_w)$ dependence was calculated from Eq.(25) with $a_w = x_w$. Note, given Eq.(20) for $\kappa_R$ the Eq.(25) can be reduced to:

$$g_{b,ideal} = \left(1 + \kappa_R \frac{a_w}{1-a_w}\right)^{1/3},\tag{30}$$

which is an analog of Eq.(19) where $\kappa = \kappa_R$.

For each experimental $g_b$ vs. $a_w$ data pairs we have calculated a $\kappa$ values using Eq.(19). Inserting fitted values of $g_b(a_w)$ into Eq.(19) we have obtained the corresponding fit curve for activity-based hygroscopicity, $\kappa$. The obtained results are shown in Fig. 10b. Due to concentration effects (Mikhailov et al., 2013 and references therein) hygroscopicity parameter decreases with $g_b$ increasing. At $a_w > 0.98$ the $\kappa$ becomes almost constant. In this area the estimated value of $\kappa$ to be $0.160 \pm 0.006$ (average of

the 10 data points ± propagated uncertainty; see insert in Fig.10b), which is close to the ideal solution value of $\kappa_R = 0.154$ (Eq.20). The hygroscopicity $\kappa$ obtained in this study for dilute glucose solution is in agreement with that derived by Ruehl et al. (2010) ($\kappa = 0.165 \pm 0.033$) measured in the 99.4-99.9 % RH range using continuous-flow thermal gradient column and with the HHTDMA - based value of $\kappa$ = 0.162 reported by Suda and Petters (2016) at RH > 90%. Note that at the same water activity range

the measurement uncertainty of $\kappa$ with the HHTDMA method is ~6 times less than that in the thermal gradient column setup (0.006 vs. 0.033). As mentioned before by Suda and Petters (2013), the optical measurements used for particles size determination (Ruehl et al., 2010; Wex et al., 2005) are subjected to limitations in accuracy resolution due to uncertainties in refractive index and the conversion from optical to physical diameter.


### 5.3.2 Activity and molal osmotic coefficients

Figure 11a shows HHTDMA-based activity coefficient of water ($\gamma_w$) and glucose ($\gamma_{Gl}$) in glucose aqueous solution. Activity coefficient of water was calculated from Eq.(15) were $x_w$ was obtained based on Eq.(8) and Eq.(9) as described in Sect.4.4, and water activity was derived from Eq.(6) with

assumptions considered in Sect. 4.2. The activity coefficient of glucose in water solution was obtained by numerical integration of Eq. (26) (Sect. 4.4).

The bulk DRH of glucose varied in the range of 88-90 % RH (Zamora et al., 2011 and references therein) that corresponds to the saturated mole fraction of glucose aqueous solution, $x_{Gl}$ of 0.095 ($\mu_{Gl}$ =3.14 mol kg$^{-1}$). Above this value, glucose particles are metastable supersaturated droplets (selected

area in Fig.11a), which are present in an amorphous (semi-solid) state. Using bulk water pressure method Taylor and Rowlinson (1955) have obtained the water activity coefficient values up to $x_w$ of 0.195 and fitted their data using two-suffix Margules Eq.(16) with $A$ = -1.957 (± 0.062). Figure 11a shows that up to $x_w$ of 0.42 our HHTDMA-derived values of $\gamma_w$ are in excellent agreement with Taylor





and Rowlinson (1955) data fit indicating that simple two-suffix Margules equation with $A = -1.957$ is
also applicable for deep metastable area. Water activity coefficients obtained in this study we also
compared with those reported by Suda and Petters (2013) (Fig.11a, green line). Their HTDMA-derived
$\gamma_w$ values are slightly lower than ours. The observed difference can be explained by the fact that Suda
and Petters (2013) used assumption of volume additivity to calculate water activity coefficient.
Moreover, as mentioned above, in their study no shape factor correction for the dry particles was made.

In addition, Fig.11a shows the glucose activity coefficients, which are compared to bulk
measurements by Miyajima et al. (1983) obtained with the isopiestic method (black symbols). One can
see that our $ln\lambda_{Gl}$ values are in a good agreement with literature data points. For future applications, we
fitted our $ln\lambda_s$ data (up to $x_w = 0.42$) together with the Miyajima et al. (1983) bulk results using a
polynomial 4th-order fit function. The obtained fitting coefficients are listed in Table 3.

Figure 11b shows HHTDMA-based molal osmotic coefficient of glucose, $\Phi_{Gl}$ as a function of
water activity. The molal osmotic coefficient was calculated from Eq. (21) where $\mu_{Gl}$ was obtained
using Eq. (14). The obtained data pairs $\Phi_{Gl}$ vs. $a_w$ were fitted using theory relation (Eq. 23) proposed
by Rudakov and Sergievski (2009) (Sect. 4.3). The best fit value of the hydration number, $h^0$ to be $1.88\pm$
$0.04$ (n = 75; $R^2 = 0.858$). That is close to $h^0 = 1.7 \pm 0.5$ reported by Rudakov and Sergievski (2009).
Our HHTDMA-based values of $\Phi_{Gl}$ are within $h^0 \pm 0.5$ (gray shaded area, Fig.11b). At $a_w > 0.98$ the
$\Phi_{Gl}$ value is $1.034 \pm 0.025$ (average ± propagated uncertainty; 11 data points). This result indicates that
even in diluted glucose solution, nonideality caused by hydration of glucose molecules still persists.
Experimental values of $\Phi_{Gl}$ we accompanied to those obtained by Suda and Petters (2013) (black circles,
Fig.10b). In general, Suda and Petters (2013) data points are close to our results. A noticeable deviation
is observed in the water activity range of $0.85 – 0.95$. The main reason is that the relatively small
changes in the instrumental uncertainties of aerosol particle growth factors and in RH will lead to large
uncertainties in the determination of their thermodynamic characteristics. Thus, for our HHTDMA
system in case of glucose aerosol particles in the RH range of 90-99% the growth factor uncertainty of
~0.6% (Eq.2) gives rise the uncertainty in $\Phi_{Gl}$ of ~3 %.

## 6. Summary and conclusions

We have demonstrated the key features of newly designed HHTDMA instrument which allows to
measure the particles hygroscopic growth with uncertainty of ~ 0.6% throughout the 2 - 99.6% RH
range. This accuracy was firstly achieved by combining the restructuring mode with conventional
hydration/dehydration mode. The tandem of two modes allowed us to minimize uncertainties associated
with morphology of the initial dry particles. Secondly, both DMAs were temperature-stabilized. The
temperature different between sheath and excess flow in DMA2 was as small as ± 0.015° C, which





made it possible to measure particle growth factors up to 99.6% RH. Throughout the all relative humidity range, the absolute RH uncertainty is less than 0.4%.

We have checked the effect of size dependence of the DMA2 transfer function width and sensitivity of the SMPS inversion algorithm on the uncertainty in particles sizing. Our test measurements have shown that effect of transfer function broadening on the particle growth factor is negligibly small. With regard to SMPS inversion algorithm and log-normal fit used for determination of the particle mobility diameter we found that particle size resolution significantly exceeds the size of individual bins. It is

possible because DMA1 selected particle are still polydisperse and a small offset voltage leads to a change in the count statistics in each size bin. As a result, the fitted size spectra and modal mobility particle diameter shifts proportionally to the voltage change. Thus, in our experiments for ~100 nm aerosol particles we were able to maintain the required initial mobility diameter with resolution of ± 0.03 nm by changing the voltage on the DMA1 rod by several tens millivolts.

Multiple experiments with h&d mode (pre-conditioning mode) have shown that this mode provides complementary information about microstructural rearrangement processes upon aerosol particles interaction with water vapor. It allowed as quantifying envelope shape and porosity of the spray-dried ammonium sulfate and glucose aerosol particles. Changing the drying conditions, we have found that in contrast to glucose aerosol particles the water release by ammonium sulfate particles is kinetically

limited most likely due to closed or partially shielded cavities. Overall, our h&d experiments showed that particle envelope and porosity are not constant. Depending on drying conditions, they can vary from case to case in a wide range of particle shape factor. Therefore, for accurate growth factor determination, we recommend combining *in-situ* restructuring mode with hydration/dehydration modes. Since the restructured particles become compact, we were able to measure the thickness of the water adsorption

layer on the surface of the ammonium sulfate particles before DRH. We found that water adsorption occurs already at lower humilities, ranging from ~15 % RH. The number monolayers linearly increased from ~0 to 3.5 for the range of 5-75 % RH and sharply increased up to ~ 6 monolayers over the range of 75-79% RH.

        Hydration/dehydration experiments with ammonium sulfate particles showed that experimental

growth factors are in a good agreement with E-AIM model confirming that after pre-conditioning the restructured particles are compact and spherical. Averaged over the range of 38-96% RH, the mean relative deviation between measurement and model results were < 0.5 %. We also tested the RH accuracy, which can be obtained from conversion of experimental growth factors into RH using E-AIM. Due to low instrumental growth factor uncertainty, we were able to measure RH above 80% with

absolute accuracy no worse than 0.3 % RH. Moreover, this uncertainty decreased with RH increasing, dropping to 0.03 % at RH = 99.5 %. Thus, using ammonium sulfate growth factors as a calibration standard it was possible to eliminate RH uncertainty generated by capacitive and dew point sensors at



RH above 80 %. In general, we have shown that tandem h&d (pre-conditioning) and hydration/dehydration modes, as well as improved methods for measuring the RH creates the
prerequisites for accurate determination of the thermodynamic parameters of aerosol particles in the wide range of RH. The effectiveness of this approach has been tested on glucose aerosol particles.

The glucose growth factors measured in the 2-99.6 % RH range are in a good agreement with literature data. At RH above 90 %, a perfect agreement between our data and those obtained by bulk methods was observed. Up to 99.6 % RH, average deviation of experimental growth factors from the
full Köhler was as small as 0.7%. At water activity above 0.98, the calculated value of κ to be $0.160 \pm 0.006$. The HHTDMA-based activity coefficient of water and glucose in glucose aqueous solution has been obtained including metastable area up to $x_w = 0.42$. Both HHTDMA-derived activity coefficients are in a good agreement with those obtained by bulk methods reported in literature. We also calculated molal osmotic coefficient of glucose and estimated hydration number, which is ~1.9. One should note
that all thermodynamic parameters were obtained without assumption of volume additivity. Since the thermodynamic characteristics of glucose aqueous solution above bulk DRH are well defined, it can also be used as a reference standard for RH determination from experimental growth factors. It will reduce the upper limit of voltage applied to DMA2 and avoid potential discharge in the column at high RH.

Overall, our results demonstrated that the HHTDMA system described in this work allows us to determine the thermodynamic characteristic of aqueous solutions with an accuracy close to that obtained by bulk methods. At the same time, an important advantage of this method is the ability to determine these characteristics for highly supersaturated solution droplets.

*Author contributions.* E.F.M. designed the study, performed the measurements, and wrote this paper. S. S. Vlasenko contributed to the discussion and interpretation of the results.

*Data availability*. Data used in this study can be made available upon request to the author.

*Competing interests*. The authors declare that they have no conflict of interest.

*Acknowledgements*. All performed studies were supported by Russian Science Foundation (grant agreement no. 18-17-00076) and Max Planck Society (MPG). We thank the Geomodel Research Center at Saint Petersburg State University.






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

940

945

950




**Table 1.** Microstructural rearrangement parameters for pure ammonium sulfate and glucose aerosol particles obtained in *h&d* experiment for different drying conditions (Fig. 4a and 4b). $D_{b,i}$ and $D_{b,h\&d,min}$ are mean ± standard deviation of 7–10 data points; dynamic shape factor, $\chi$ , porosity, $\delta$ , and void fraction, $f$ together with propagated error, $\Delta f$ are calculated from Eq.(3), Eq.(4), and Eq.(5) respectively. r.t. is the residence time. The $\chi$ and $\delta$ propagated uncertainty is better than ± 0.002.

| Type of dryer | r.t. (s) | $D_{b,i}$ (nm) | $D_{b,h\&d,min}$ (nm) | $\chi$ | $\delta$ | $f \pm \Delta f$ (%) |
|---|---|---|---|---|---|---|
| Ammonium sulfate | | | | | | |
| MD-700 | 27 | 100.26 ± 0.03 | 98.39 ± 0.03 | 1.033 | | |
| MD-700 | 41 | 100.26 ± 0.05 | 98.21 ± 0.03 | 1.037 | | |
| MD-700 | 54 | 100.26 ± 0.02 | 98.15 ± 0.05 | 1.038 | | |
| MD-700 | 67 | 100.25 ± 0.02 | 98.00 ± 0.05 | 1.040 | | |
| MD-700 + SDD | 27 + 62 | 100.24 ± 0.03 | 95.32 ± 0.03 | 1.092 | 1.032 | 9.0 ± 0.3 |
| | | 100.26 ± 0.02 | 95.39 ± 0.03 | 1.090 | 1.031 | 8.7 ± 0.3 |
| SDD | 62 | 100.26 ± 0.03 | 93.16 ± 0.04 | 1.137 | 1.056 | 15.1 ± 0.2 |
| | | 100.26 ± 0.02 | 93.45 ± 0.06 | 1.131 | 1.053 | 14.3 ± 0.2 |
| Glucose | | | | | | |
| MD-700 | 27 | 100.25 ± 0.03 | 96.94 ± 0.03 | 1.060 | 1.034 | 9.5 ± 0.3 |
| MD-700 | 41 | 100.24 ± 0.01 | 96.74 ± 0.03 | 1.064 | 1.036 | 10.2 ± 0.3 |
| MD-700 + SDD | 41 + 62 | 100.25 ± 0.05 | 96.7 ± 0.04 | 1.064 | 1.036 | 10.1 ± 0.3 |
| SDD | 62 | 100.25 ± 0.04 | 96.96 ± 0.04 | 1.060 | 1.034 | 9.5 ± 0.3 |

**Table 2.** Parameters characterizing the hygroscopic properties of glucose aerosol particles: best-fit values (± standard errors) for the three-parameter fit ($k_1$, $k_2$, $k_3$; Eq.17). $n$ and $R^2$ are the number of data points and the coefficient determination of the fit, respectively.

| $k_1$ | $k_2$ | $k_3$ | $R^2$ | $n$ | $a_w$ range |
|---|---|---|---|---|---|
| 0.2629 ± 0.0272 | 0.05796 ± 0.0662 | -0.1655 ± 0.0399 | 0.9994 | 142 | 0.02 - 0.98 |

**Table 3.** Fitted parameters (± standard deviation) of $ln\lambda_s$ as a function of mole fraction of glucose, $x_{Gl}$ in glucose aqueous solution. $n$ and $R^2$ are the number of data points and the coefficient determination of the fit, respectively.

| Polynomial fit function: $ln\gamma_{Gl} = B_0 + B_1 x_{gl} + B_2 x_{Gl}^2 + B_3 x_{Gl}^3 + B_4 x_{Gl}^4$ | | | | | | | |
|---|---|---|---|---|---|---|---|
| $B_0$ | $B_1$ | $B_2$ | $B_3$ | $B_4$ | $n$ | $R^2$ | $x_w$ range |
| -0.0085 ± 0.0059 | 2.5846 ± 0.3158 | 17.880 ± 3.675 | -79.036 ± 14.862 | 92.138 ± 19.152 | 103 | 0.996 | 0.002 - 0.42 |

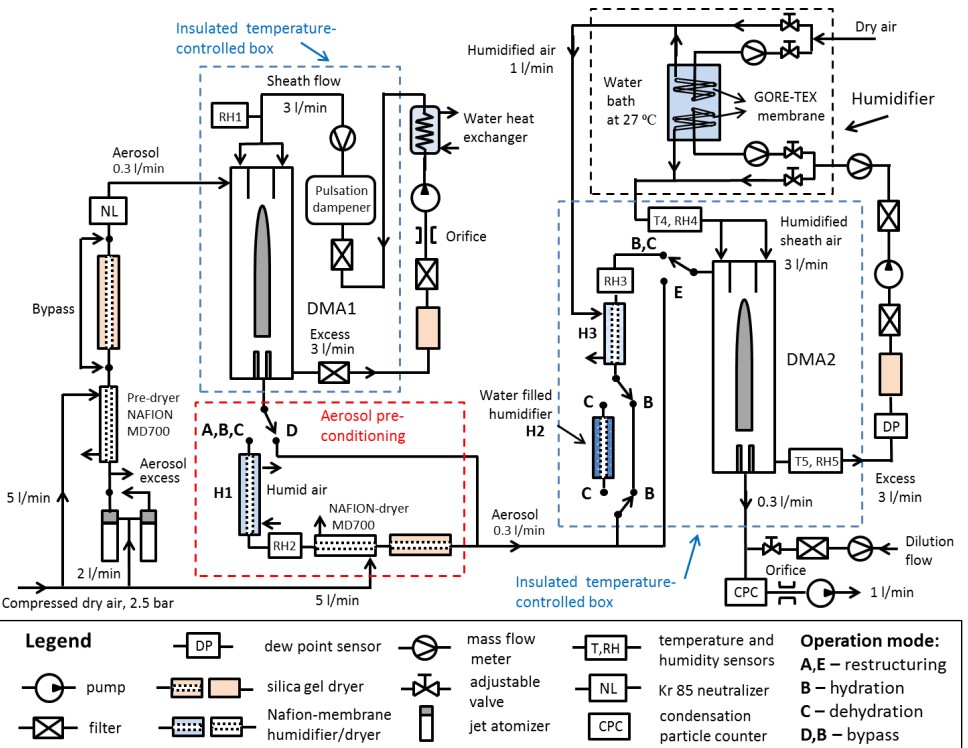

**Fig. 1.** Schematic design of HTDMA setup: RH–RH5 – relative humidity sensors (Almemo, FHAD
46C41A); T4, T5 – needle sensors (Pt100, 1/3, 300×1.5 mm, DOSTMANN-electronic); DP – dew
point sensor (Dew Master, Edgetech Instrument, remote D-probe SC); DMA1 DMA2 – differential
mobility analyzer (TSI 3081), mass flow meter –  (TSI 4040), NAFION  humidifier (Perma Pure;
MD-110/P), jet atomizer – (3076, TSI), CPC – condensation particle counter (3772, TSI)

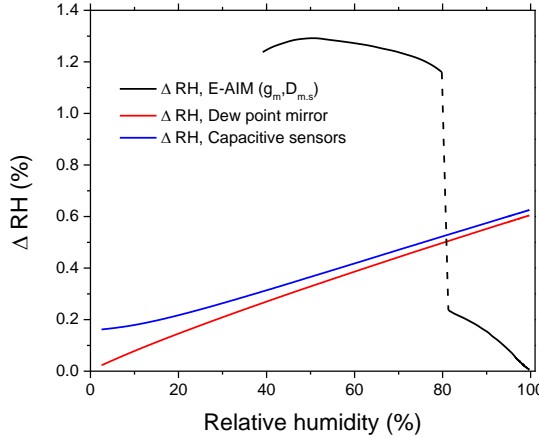

**Fig. 2.** Accuracy in RH using different methods.






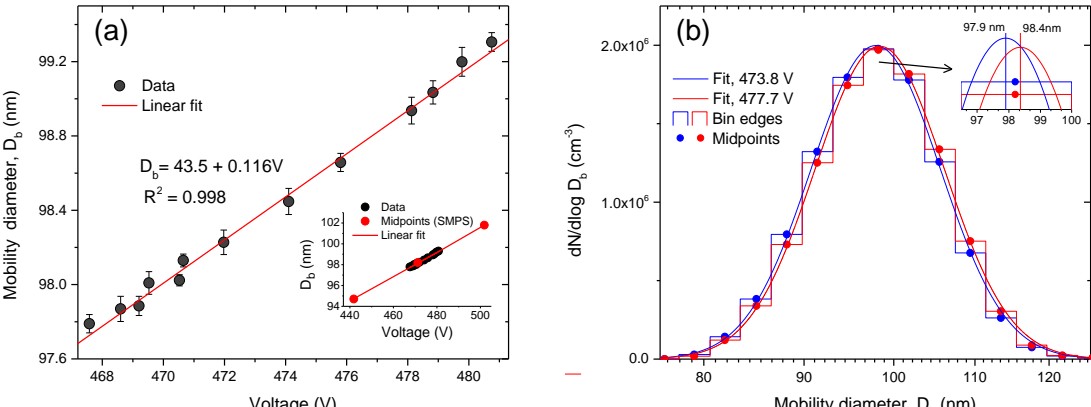

**Fig.3** Modal mobility equivalent diameters obtained by the log-normal fit of the SMPS size distribution (DMA2) as a function of the voltage applied to DMA1 center rod **(a)** and histogram together with fit curve received for two selected voltage **(b)**.


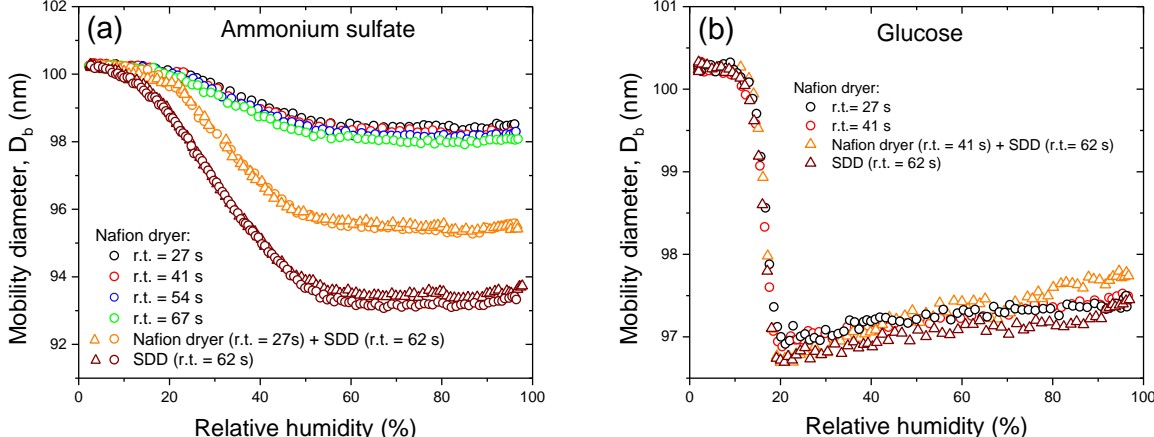

**Fig. 4** Mobility equivalent diameters of ammonium sulfate **(a)** and glucose **(b)** with the initial dry mobility equivalent diameter, $D_{b,i} = 100.3$ nm observed upon hydration & dehydration (h&d mode, RH2) depending on drying conditions. Different symbols are different experimental runs (panel **a**).


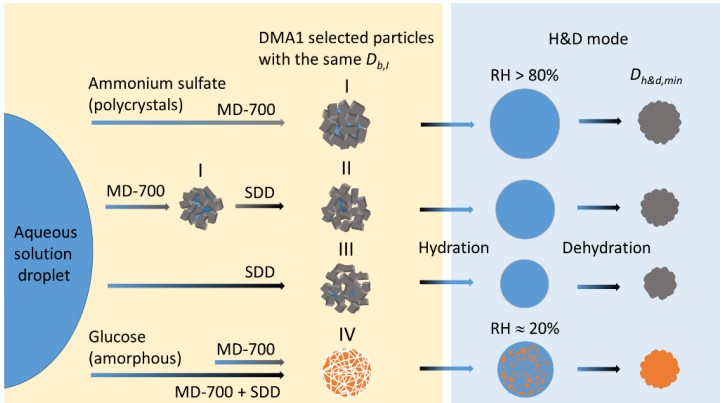


**Fig. 5** Possible structures of polycrystalline ammonium sulfate and amorphous glucose aerosol particles depending on drying conditions: (I) agglomerate of single crystals with fully and partly shielded cavities filled with liquid; (II) and (III) polycrystalline agglomerates with open and shielded

cavities, having different void to solid ratio; (IV) amorphous glucose aerosol particle in gel-like state. Other explanations are given in the text.

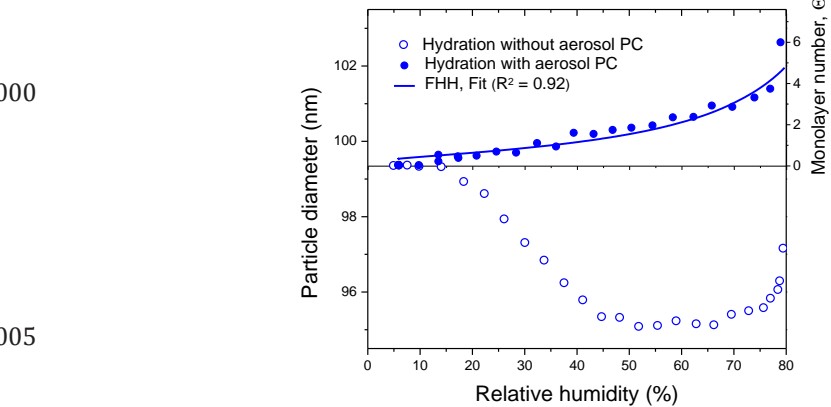



**Fig. 6** Mobility equivalent diameters of ammonium sulfate particles observed in hydration

experiments with and without pre-conditioning and equivalent number of monomolecular layers. $\Theta$ was calculated from Eq.(27) assuming that $D_{b,h\&d,min} = D_{m.s} = 99.35 \pm 0.03$ nm, obtained with aerosol particle pre-conditioning. Line is Köhler model fit, (Eq. 6) with water activity from FHH adsorption isotherm (Eq. 28).





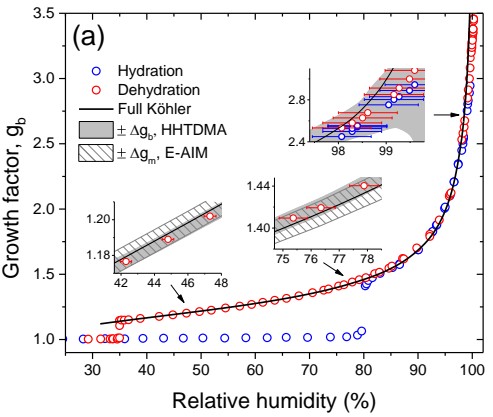

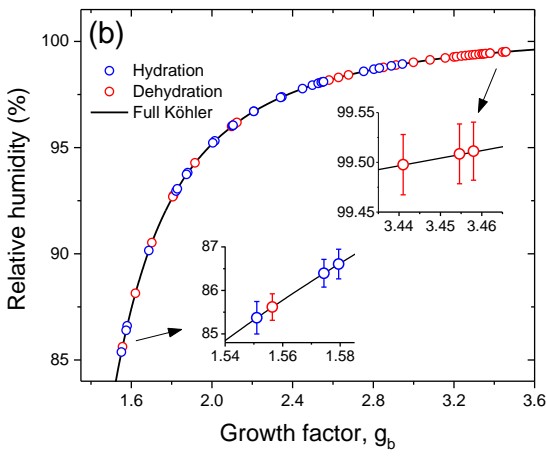


**Fig. 7.** Growth factors observed in hydration and dehydration experiments of the pre-conditioned ammonium sulfate aerosol particles with $D_{b.h\&d,min} = 79.6$ nm as compared with full Köhler model: (**a**) growth factors as a function of relative humidity measured with capacitive RH probe (RH4, Fig.1); (**b**) RH values were obtained from E-AIM using experimental growth factors. Insets in panel (**a**): the gray area denotes the growth factor
uncertainty obtained from Eq. (2); the shaded area corresponds to growth factor uncertainty of E-AIM below DRH obtained from EDB experimental data. Whiskers show RH uncertainty (panels **a**,**b**).

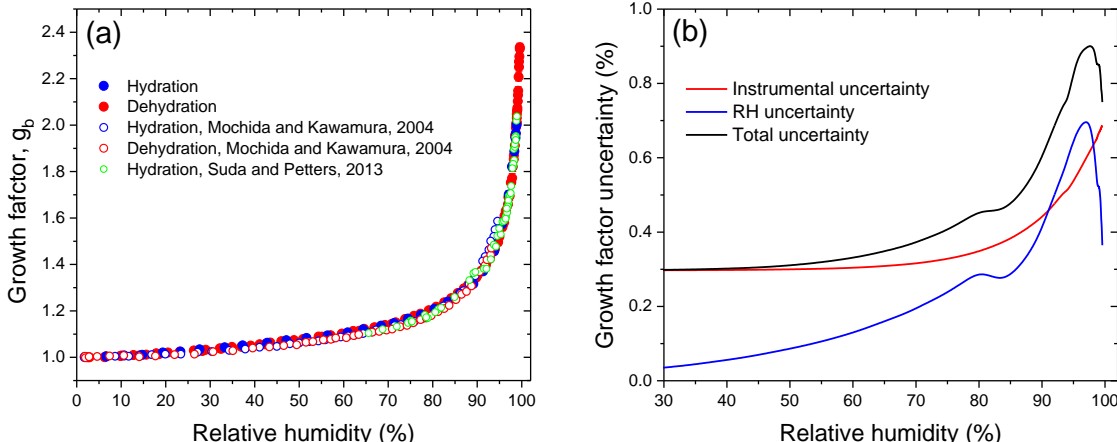

**Fig. 8**. Growth factors observed in hydration and dehydration experiments of the pre-conditioned glucose aerosol particles with $D_{b.h\&d,min} = 99.6$ nm in comparison with literature data (**a**) and relative growth factor uncertainty due to instrumental and RH errors (**b**).






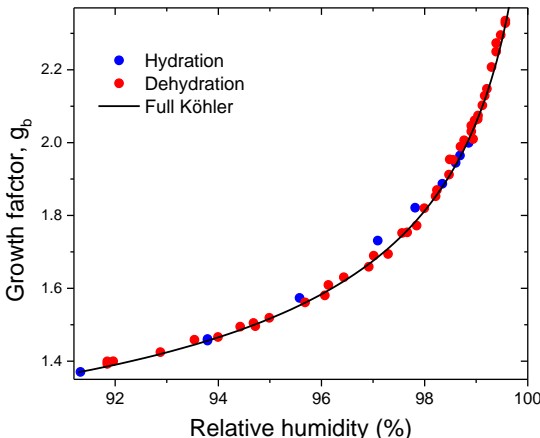

**Fig. 9**. Growth factors observed in hydration and dehydration experiments of the pre-conditioned glucose aerosol particles with $D_{b.h\&d,min}$ = 99.6 nm in comparison to mass equivalent growth factors calculated with full Köhler model.


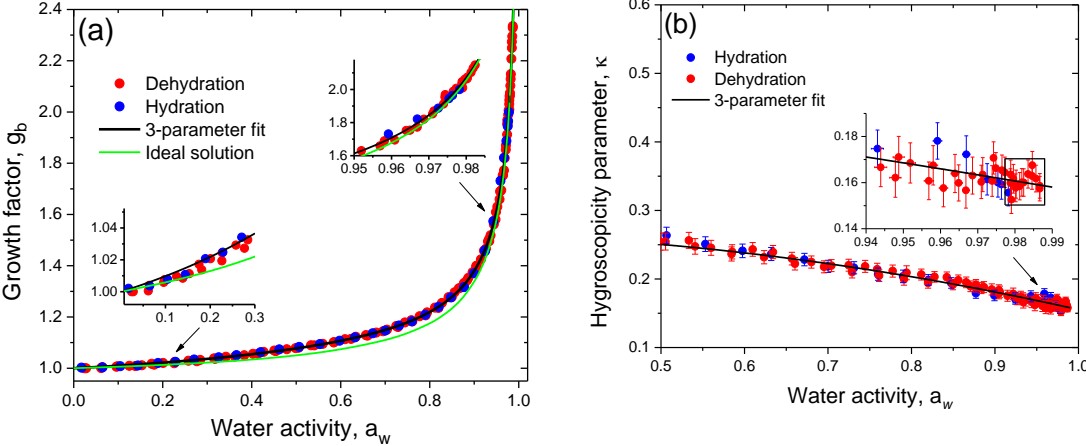

**Fig. 10**. Growth factor (**a**) and hygroscopicity parameter (**b**) of glucose aerosol particles as a function of water activity. Black lines in panels (**a**) and (**b**) correspond to three-parameter fit using Eq.(17) and Eq.(19), respectively. Green line account for ideal solution model (**a**). Inserts: (**a**) show water activity based growth factors at low and high $a_w$ in comparison with model curves; (**b**) shows hygroscopicity parameter change at $a_w$ above 0.94; data points selected by the rectangle are used to calculate the average value of dilute hygroscopicity parameter, $\kappa$.






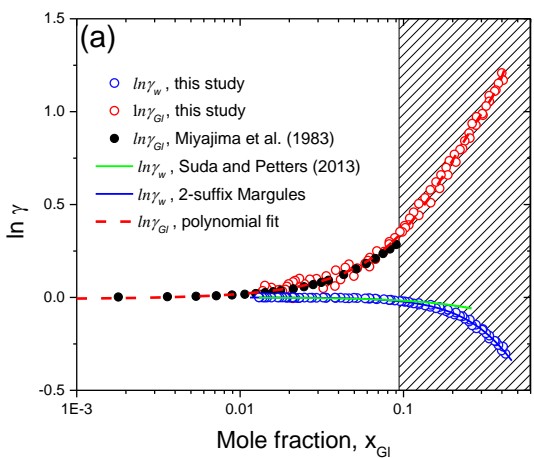
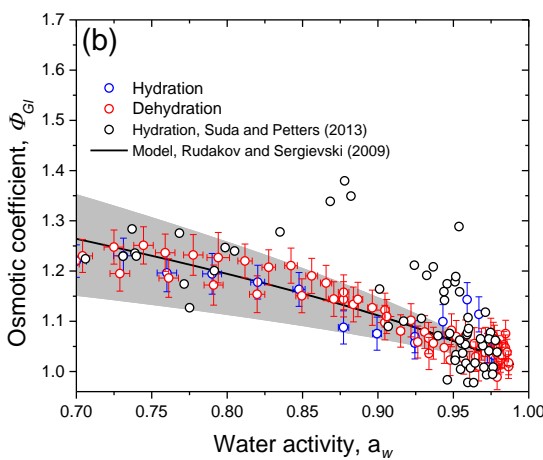


**Fig.11**. HHTDMA-based activity coefficients of water ($\gamma_w$) and glucose ($\gamma_{Gl}$) as a function of mole fraction (**a**) and molal osmotic coefficient of glucose ($\Phi_{Gl}$) vs. water activity (**b**) for glucose solution droplets in comparison with literature data. Bulk measurement of $\gamma_{Gl}$ from Miajima et al., (1983) – black solid (**a**); the data points and error bars are from HHTDMA experiment of hydration (blue circles) and dehydration (red circles) (**b**), $\Phi_{Gl}$ from Suda and Petters (2013) – black circles (**b**). Model lines: (**a**) 2-suffix Margules equation (Eq.16, with $A = -1.957$) – blue solid; (**b**) Rudakov and Sergievski (2009) model (Eq. 23) with hydration number of $h^0 = 1.88$ (the best fit parameter with standard error is of 0.04) – black solid. Gray shaded area denotes hydration number range with the $h^0 + 0.5$ (top bound) and $h^0 - 0.5$ (low bound). Red dashed fit line in panel (**a**) is the polynomial 4th-order fit function of $ln\gamma_{Gl}$ obtained in this study together with Miajima et al. (1983) bulk measurements. The shaded rectangle in panel (**a**) denotes metastable area of glucose solution droplets.

