# Peer review of "High humidity tandem differential mobility analyzer for accurate determination of aerosol hygroscopic growth, microstructure and activity coefficients over a wide range of relative humidity"

_Atmospheric Measurement Techniques, 2019_

## Referee Comment (RC1) · Anonymous Referee #1 · 8 Jan 2020

General comments:

This paper presents a description of a HHTDMA and related method developed for investigating the hygroscopicity of aerosols in RH range of 2-99.6% with the uncertainty of growth factors within 0.9%, which will help explore the interaction between water and aerosols at RH close to 100%. By combining the restructuring modes with hydration/dehydration modes, the GFs can be measured in high precision after the microstructural rearrangement effect is considered. The manuscript is well written and presents a valuable contribution in the field of aerosol measurement techniques. I recommend this manuscript to be published after the following issues to be addressed and modified.

Specific comments:

Section 2.1: The manuscript gives a general description about the design and the components in constructing the HHTDMA. However, temperature and humidity control should be critical issues in operation, for example, maybe a PID control program was used with the input of RH4 and RH5 probe to control the RH in DMA2 precisely. How did the GORE-TEX membrane work? How to adjust the rotation speed of the fans in the DMA2 box? Has the temperature gradient in DMA2 column been estimated? More details about the humidity and temperature control of the second DMA would be better.

Page 13, Line 358-360: This sentence is obscure and should be rewritten.

Page 16, Line 470-480: I recommend providing more proofs (e.g. SEM images of particles) or reference(s) to support this claim.

---

## Referee Comment (RC2) · Anonymous Referee #2 · 15 Feb 2020

Mikhailov et al. present an instrument characterization of a newly constructed high humidity tandem DMA instrument. The instrument shows improved capabilities compared to previously described setups. The manuscript is well written and I recommend it for publication in AMT.

Minor comments

The authors might consider adding a section comparing the versatility and accuracy of

the system with other techniques. Specifically the Leipzig based LACIS instrument and the filter-based mass-based hygroscopicity method used by the same author previously would be interesting to compare in this context.

The authors state that activity coefficients can be determined from the data without the need to assume volume additivity by relying only on known (bulk) solution density. The authors should add that solution density is rarely known for metastable solutions and systems of interest to be studied with the HHTDMA.

The data should be made available in a FAIR aligned repository. Making data "available upon request to the author" is inconsistent with the AMT data policy (https://www.atmospheric-measurement-techniques.net/about/data_policy.html).

―――――――――――――――――――――

---

## Referee Comment (RC3) · Anonymous Referee #3 · 17 Feb 2020

Summary:

The manuscript "High humidity tandem differential mobility analyser for accurate determination of aerosol hygroscopic growth, microstructure and activity coefficients over a wide range of relative humidity" presents a revised and improved instrument design to cover the relative humidity range from 2-99.6% for hygroscopicity measurements with high measurement accuracy. The instrument is further used to study ammonium

sulphate and glucose particles and results are compared to former literature results and model studies. Particularly, the effect of aerosol restructuring is investigated and discussed to improve the analysis of the water uptake. I believe that the manuscript is very interesting and well written and I therefore recommend the paper for publication after the following minor comments have been addressed:

Minor comments:

I would strongly recommend revising the use of the English language throughout the manuscript as many small mistakes (see specific comments) are currently present and some sentences could profit from being re-written for a better understanding.

Specific comments:

Page 2, line 52: replace "result" with "resulting"

Page 2, line 63: replace "due" with "to"

Page 2, line 67: add missing parenthesis after the citations

Page 2, line 68: replace "effect" with "effects"

Page 3, line 76: rephrase the sentence starting with "However, due to. . .."

Page 3, line 82: add "for" after "this instrument allows"

Page 3, line 88: rephrase the sentence starting with "The averaged in the 80-99%..."

Page 4, line 110: add "a" in front of "circulation thermostat"

Page 4, line 111: add "are" before "operated at 26C.."

Page 4, line 114 and following: I would suggest to use present tense and not past tense to describe the difference in e.g. PT100 sensors uncertainty.

Page 5, line 145: replace "nebulize" by "nebulizing"

Page 5, line 162: the units of 1 l/m should rather be 1 l/min

Page 5, line 172: rephrase the whole sentence, very unclear

Page 7, line 214: wrong units for the error of the dry mobility diameter (should be nm)

Please check carefully the whole manuscript for missing articles like "the" or "a", singular and plurals and the use of past and present tense.
* * *

---

## Author Comment (AC1) · 12 Mar 2020

Response to Anonymous Referee #1

The referee's comments are in italics, our responses in plain font.

*This paper presents a description of a HHTDMA and related method developed for investigating the hygroscopicity of aerosols in RH range of 2-99.6% with the uncertainty of growth factors within 0.9%, which will help explore the interaction between water and aerosols at RH close to 100%. By combining the restructuring modes with hydration/dehydration modes, the GFs can be measured in high precision after the microstructural rearrangement effect is considered. The manuscript is well written and presents a valuable contribution in the field of aerosol measurement techniques. I recommend this manuscript to be published after the following issues to be addressed and modified.*

We thank the Referee #1 for the suggestions for improvement that were taken into account upon manuscript revision. Responses to individual comments are given below.

*Specific comments:*
*Section 2.1: The manuscript gives a general description about the design and the components in constructing the HHTDMA. However, temperature and humidity control should be critical issues in operation, for example, maybe a PID control program was used with the input of RH4 and RH5 probe to control the RH in DMA2 precisely.*

The temperature and RH control is discussed in Sect. 2.1 and Sect 2.4, respectively. PID control program was not used. The temperature **gradient**, i.e. the temperature profile along the DMA2 column (dT/dL) was not directly measured. The temperature difference between sheath and excess flow was used to estimate temperature variation inside DMA2 as indicated in Sect. 2.1.

*How did the GORE-TEX membrane work?*

The following clarifying text has been added in Sect. 2.3 to explain how Gore-Tex membrane was used:
The humidity of the aerosol flow (RH3) and sheath air (RH4) in DMA2, is controlled by mixing water saturated and dry air flows in a ratio produced the desired RH. Saturated air is obtained by passing dry air through a Gore-Tex membrane tube submerged inside a temperature controlled water bath ($27.0 \pm 0.1$ ºC). Two separate 6 mm (ID) Gore-Tex tubes, 0.5-m and 2-m long are used for aerosol and sheath flows conditioning, respectively (Humidifier, Fig. 1). For the H1 Nafion exchanger the humid air is prepared by bubbling air directly through water and then mixing with dry air to the required humidity (not shown in Fig.1).

*How to adjust the rotation speed of the fans in the DMA2 box?*

The fan speed can be changed by varying the applied voltage (manually), but this was not necessary. In Supp. 2.1 we have demonstrated a simple way to compensate for the temperature difference between sheath and excess flows, if it needed.
The text in Suppl. 2.1 was modified as following:
The test measurements showed that the temperature difference between the sheath and excess flows can be changed within $\pm0.3$ ºC by adjusting the rotation speed of the fans. The speed of each fan is affected by applied AC voltage.

*Page 13, Line 358-360: This sentence is obscure and should be rewritten.*

The sentence is updated:

The FHH (Frenkel, Halsey and Hiil) model is the frequently used to relate surface coverage to a water activity:

$$a_w = exp(-A_{FHH}/\Theta^{B_{FHH}}), \tag{28}$$

where A$_{FHH}$ and B$_{FHH}$ are empirical fit parameters that describe the intermolecular interactions governing the adsorption potential. $A_{FHH}$ characterizes interactions between the surface and first adsorbed water layer as well as interactions between adjacent molecules. $B_{FHH}$ describes the interactions between the surface and subsequent adsorbate layers.

*Page 16, Line 470-480: I recommend providing more proofs (e.g. SEM images of particles) or reference(s) to support this claim.*

The conclusion about the difference in the porous structure of ammonium sulfate and glucose particles is based on the results of HHTDMA measurements in the h&d mode. To our knowledge, there are no direct methods for measuring the pore network of 100 nm particles. SEM is a useful technique for extracting two-dimensional (2D) images of the microstructures but does not provide the third spatial component of the sample, which is important to find interconnected regions and pore volumes, shapes and sizes.
We have added the SEM images of initial ammonium sulfate and glucose aerosol particles (Fig. 5) in order to strengthen the argument in favor of the discussed aerosol particles morphology and the calculated values of particle shape ($\chi$, $\beta$) and porosity ($\delta$, $f$) presented in Table 1.

[Figure]

**Fig. 5** SEM images of initial ammonium sulfate (A) and glucose (B) aerosol particles. The samples were investigated with a high-resolution SEM (ZEISS Merlin). Operation conditions: 0.4 kV accelerating voltage, 1.5 kV ESB grid voltage, 1.8 mm working distance. Particle samples were collected directly onto a 3mm TEM copper 300 mesh grids, coated with a 30–60 nm thick Formvar film.

---

## Author Comment (AC2) · 12 Mar 2020

Response to Anonymous Referee #2.

The referee's comments are in italics, our responses in plain font.

Mikhailov et al. present an instrument characterization of a newly constructed high humidity tandem DMA instrument. The instrument shows improved capabilities compared to previously described setups. The manuscript is well written and I recommend it for publication in AMT.

We thank the Referee #2 for these positive remarks. Responses to individual comments are given below.

The authors might consider adding a section comparing the versatility and accuracy of the system with other techniques. Specifically the Leipzig based LACIS instrument and the filter-based massbased hygroscopicity method used by the same author previously would be interesting to compare in this context.

**The following text has been added:**

In addition to the HTDMA methods, other techniques have been used to determine the aerosol hygroscopicity at high RH (Tang et al., 2019). Two of these methods are the Leipzig Aerosol Cloud Interaction Simulator (LACIS; Stratmann et al., 2004) and the inverted streamwise-gradient cloud condensation nuclei counter (Ruehl et al. 2010), which could be operated at RH over the range of 85.8 - 99.1 % and 99.4 - 99.9 %, respectively. Both methods have accurate humidity control, but the optical detectors used to determine the wet particle size distribution are subjected to limitations in accuracy resolution due to uncertainties in refractive index and the conversion from optical to physical diameter. This leads to uncertainty in the measured growth factors of ~ 4% (Wex et al., 2005).

Mikhailov et al. (2011) developed a filter-based differential hygroscopicity analyzer (FDHA), which was employed as an offline method to investigate hygroscopic properties of ambient aerosol particles (Mikhailov et al., 2013, 2015). An updated version of the instrument allows measuring the hygroscopic growth up to 99.6 % with accuracy of  $\pm$  0.1 % RH. The uncertainty in the determination of the mass growth factors was estimated to be ~1 % at 30 % RH and ~10% at 99 % RH. FDHA measures water mass absorbed by aerosol particles deposited on the filters. Due to mass conservation, this method is not influenced by the effects of capillary condensation and restructuring of porous and irregularly shaped particles that usually limit the applicability and precision of mobility diameter-based HTDMA and CCNC (Cloud Condensation Nuclei Counter) experiments. Since FDHA is katharometer-based technique, it takes on average of 2 days, to measure one aerosol sample, which is a drawback of this instrument.

The authors state that activity coefficients can be determined from the data without the need to assume volume additivity by relying only on known (bulk) solution density. The authors should add that solution density is rarely known for metastable solutions and systems of interest to be studied with the HHTDMA.

On line 338 the following text has been added:

For many atmospheric aerosols, the concentration dependence of the aqueous solution density is not well defined. At the same time, for a number of model systems of interest, the aerosol solution density was measured in both unsaturated and supersaturated solutions. In this case  $x_w$  can be obtained without assumption of volume additivity by iteratively solving Eq. (8) with other equation where  $\rho$  and concentration is given explicitly.

The data should be made available in a FAIR aligned repository. Making data "available upon request to the author" is inconsistent with the AMT data policy (https://www.atmospheric-measurement-techniques.net/about/data\_policy.html).

The data are available at <a href="https://osf.io/87526/">https://osf.io/87526/</a>

---

## Author Comment (AC3) · 12 Mar 2020

Response to Anonymous Referee #3.

The referee's comments are in italics, our responses in plain font.

*Mikhailov et al. present an instrument characterization of a newly constructed high humidity tandem DMA instrument. The instrument shows improved capabilities compared to previously described setups. The manuscript is well written and I recommend it for publication in AMT.*

We thank the Referee #3 for these positive remarks.

*Specific comments:*

*Page 2, line 52: replace "result" with "resulting"*
Corrected
*Page 2, line 63: replace "due" with "to"*
Corrected
*Page 2, line 67: add missing parenthesis after the citations*
Corrected
*Page 2, line 68: replace "effect" with "effects"*
*Corrected*
*Page 3, line 76: rephrase the sentence starting with "However, due to: : :."*

A new version is:
However, the resulting growth factor error at high humidity is significant since the relative humidity was obtained using a dew point sensor. (Further down in the text.) Thus, at RH = 97.7 % the precision quoted by authors in absolute units is ±1.2 % and particle growth factor uncertainty is 16.6 % (±0.46 at growth factor value of 2.79).

*Page 3, line 82: add "for" after "this instrument allows"*
Corrected
*Page 3, line 88: rephrase the sentence starting with "The averaged in the 80-99%..."*
The sentence was modified as following:

The resulting growth factor uncertainty associated with RH and instrumental errors is ~2%, which is propagated in hygroscopicity and activity coefficients of ±20 %.

*Page 4, line 110: add "a" in front of "circulation thermostat"*
Corrected
*Page 4, line 111: add "are" before "operated at 26C.."*
Corrected
*Page 4, line 114 and following: I would suggest to use present tense and not past tense to describe the difference in e.g. PT100 sensors uncertainty.*
Corrected
*Page 5, line 145: replace "nebulize" by "nebulizing"*
Corrected
*Page 5, line 162: the units of 1 l/m should rather be 1 l/min*
Corrected
Page 5, line 172: rephrase the whole sentence, very unclear

The text was modified as following:

The humidity of the aerosol flow (RH3) and sheath air (RH4) in DMA2, is controlled by mixing water saturated and dry air flows in a ratio produced the desired RH. Saturated air is obtained by passing dry air through a Gore-Tex membrane tube submerged inside a temperature controlled water bath (27.0 ± 0.1 ºC). Two separate  6 mm (ID)  Gore-Tex tubes,  0.5-m and 2-m long are used for aerosol and sheath flows conditioning, respectively (Humidifier, Fig. 1). For the H1 Nafion

exchanger the humid air is prepared by bubbling air directly through water and then mixing with dry air to the required humidity (not shown in Fig.1).

*Page 7, line 214: wrong units for the error of the dry mobility diameter (should be nm)*
Corrected
*Please check carefully the whole manuscript for missing articles like "the" or "a", singular and plurals and the use of past and present tense.*

Done.